# MULTI-AGENT POLICY OPTIMIZATION WITH APPROXIMATIVELY SYNCHRONOUS ADVANTAGE ESTIMATION

## ABSTRACT

Cooperative multi-agent tasks require agents to deduce their own contributions with shared global rewards, known as the challenge of credit assignment. General methods for policy based multi-agent reinforcement learning to solve the challenge introduce differentiate value functions or advantage functions for individual agents. In multi-agent system, polices of different agents need to be evaluated jointly. In order to update polices synchronously, such value functions or advantage functions also need synchronous evaluation. However, in current methods, value functions or advantage functions use counter-factual joint actions which are evaluated asynchronously, thus suffer from natural estimation bias. In this work, we propose the approximatively synchronous advantage estimation. We first derive the marginal advantage function, an expansion from single-agent advantage function to multi-agent system. Further more, we introduce a policy approximation for synchronous advantage estimation, and break down the multi-agent policy optimization problem into multiple sub-problems of single-agent policy optimization. Our method is compared with baseline algorithms on StarCraft multi-agent challenges, and shows the best performance on most of the tasks.

## 1 INTRODUCTION

Reinforcement learning(RL) algorithms have shown amazing performance on many single-agent(SA) environment tasks (Mnih et al., 2013)(Jaderberg et al., 2016)(Oh et al., 2018). However, for many real-world problems, the environment is much more complex where RL agents often need to cooperate with other agents. For example, taxi scheduling(Nguyen et al., 2018) and network control(Chu et al., 2019).

In cooperative multi-agent tasks, each agent is treated as an independent decision-maker, but can be trained together to learn cooperation. The common goal is to maximize the global return in the perspective of a team of agents. To deal with such tasks, the architecture of centralized training and decentralized executions(CTDE) is proposed(Oliehoek & Vlassis, 2007)(Jorge et al., 2016). The basic idea of CTDE is to construct a centralized policy evaluator, which only works during training and is accessible to global information. At the same time, each agent is assigned with a local policy for decentralized execution. The role of the evaluator is to evaluate agents' local policies differentially from the global perspective.

A challenge in construction of centralized evaluator is multi-agent credit assignment(Chang et al., 2004): in cooperative settings, joint actions typically generate only global rewards, making it difficult for each agent to deduce its own contribution to the team's success. Credit assignment requires differentiate evaluation for agents' local policies, but designing individual reward function for each agent is often complicated and lacks of generalization(Grzes, 2017)(Mannion et al., 2018). Current policy based MARL methods generally realize credit assignment by introducing differentiate value functions or advantage functions(Foerster et al., 2018)(Lowe et al., 2017). However, these value functions or advantage functions are estimated asynchronously but decentralized policies are updated synchronously, as shown in figure 1(b), which results in natural estimation bias.

In this paper, we propose a novel policy based MARL method called multi-agent policy optimization with approximatively synchronous advantage estimation(ASAE). In our work, we first define the counter-factual scenes, in which MA advantage estimation can be converted to SA advantage estimation. For certain agent, each counter-factual scene is assigned with a SA advantage. Then

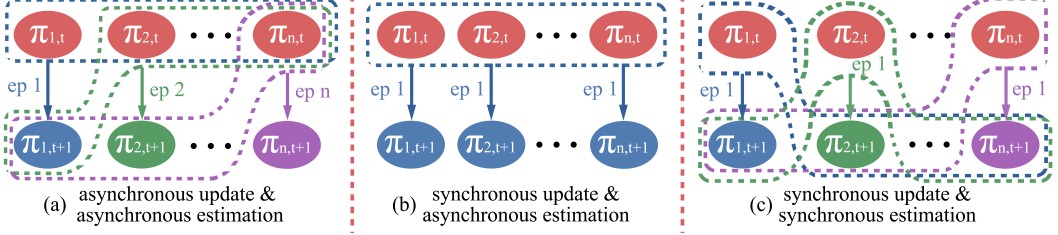

Figure 1: Comparison among three different manners of advantage estimation & update. $\pi_{a,t}$ represents the policy of agent $a$ at iteration $t$ and there are $n$ agents in total. Lines with arrow represent policy updates and $ep$ denotes the update epoch. In single iteration, synchronous update takes only one epoch while asynchronous update takes $n$ epochs. In advantage estimation, policies need to be estimated jointly, and the dashed boxes contain joint polices used for advantage estimation in corresponding update. Particularly, synchronous estimation requires other agents' future polices.

the marginal advantage function is defined as the expectation of SA advantages on distribution of counter-factual scenes, and credit assignment is realized by constructing different scenes' distribution for different agents. Moreover, in order to achieve synchronous advantage estimation, an approximation of other agents' joint future policy is introduced. To ensure the approximation is reliable, a restriction is applied to the original multi-agent policy optimization(MAPO) problem. The approximate optimization problem is simplified and broken down into multiple sub-problems, which has a similar form to trust region policy optimization(TRPO) problem. And the sub-problems are finally solved by proximal policy optimization(PPO) method.

We have two contributions in this work: (1) A novel advantage estimation method called marginal advantage estimation, which realizes credit assignment for MARL is proposed. More importantly, this method provides a channel for various SA advantage functions expanding to multi-agent system. (2) A simple yet effective method for approximatively synchronous advantage estimation is firstly proposed.

## 2 RELATED WORK

A common challenge in cooperative multi-agent tasks is credit assignment. RL algorithms designed for single-agent tasks, ignore credit assignment and take other agents as part of partial observable environment. Such algorithms perform poorly in complex cooperative tasks which require high coordination(Lowe et al., 2017). To deal with the challenge, some value based MARL methods estimate a local $Q$ value for each agent, and the shared global $Q$ value is then constructed through these local $Q$ values. Value decomposition network(VDN) constructs the global $Q$ value by simply adding all local $Q$ values together(Sunehag et al., 2018). And in QMIX algorithm(Rashid et al., 2018), the global $Q$ value is obtained by mixing local $Q$ values with a neural network. In mean field multi-agent methods, local $Q$ values are defined on agent pairs. The mapping from local $Q$ values to the global $Q$ value is established by measuring the influence of each agent pair's joint action to the global return(Yang et al., 2018).

Similarly, for policy based MARL methods, credit assignment is generally realized through differentiated evaluation with CTED structure. Some naive policy based methods estimate local $Q$ values for individual agents with a centralized critic(Lowe et al., 2017), resulting in large variance. Some other methods try to introduce advantage function in MARL. Counter-factual multi-agent policy gradient(COMA) method(Foerster et al., 2018) is inspired by the idea of difference reward(Wolpert & Tumer, 2002) and provides a naive yet effective approach for differentiated advantage estimation in cooperative MARL. In COMA, a centralized critic is used to predict the joint $Q$ value function $Q^\pi(s, \mathbf{u})$ of joint action $\mathbf{u}$ under state $s$. And the advantage for agent $a$ is defined as

$$A^a(s, \mathbf{u}) = Q(s, \mathbf{u}) - \sum_{u'^a} \pi^a(u'^a | \tau^a) Q(s, (u^{-a}, u'^a)) \tag{1}$$

where $\tau$ and $\pi$ represent trajectory and policy respectively. $a$ and $\mathbf{-a}$ denote current agent and the set of other agents respectively. COMA introduces a counter-factual baseline, which assumes that

other agents take fixed actions, as shown in figure 1(b). COMA performs synchronous updates with asynchronous estimation, which leads to lagging and biased advantage estimation. In contrast, asynchronous estimation & asynchronous updating is more reliable yet more complicated. An ideal approach is synchronous estimation & synchronous updating. However, it requires prediction of other agents' future policies.

## 3 BACKGROUND

We consider a most general setting of partially observable, full cooperative multi-agent tasks, which can be described as a stochastic game defined by a tuple $G = < S, U, P, r, Z, O, n, \gamma >$. The true state of environment $s \in S$ is unavailable to all agents. At each time step, $n$ agents identified by $a \in A$ ($A = \{1, 2, \cdots, n\}$) receive their local observations $z^a \in Z$, and take actions $u^a \in U$ simultaneously. The joint observation $\mathbf{Z} = Z^n$ is acquired by the observation function $O(s, a)$ : $S \times A \to \mathbf{Z}$. The next state is determined by joint action $\mathbf{u} \in \mathbf{U}$ ($\mathbf{U} = U^n$) and the transition function $P(s'|s, \mathbf{u}) : S \times \mathbf{U} \times S \to [0, 1]$. The reward function $r(s, \mathbf{u}) : S \times \mathbf{U} \to \mathbb{R}$ is shared by all agents, so as the discounted return $G_t = \sum_{t+i}^{\infty} \gamma^t r_{t+i}$. $\gamma \in [0, 1)$ is a discount factor.

In policy based MARL with CTED architecture, each agent has a local trajectory $\tau^a$ consists of historical observation and action $\{(z_0^a, u_0^a), (u_1^a, z_1^a), \cdots\}$. And an independent policy $\pi^a(u^a|\tau^a)$ is constructed for each agent on their local trajectory. Action-state value function $Q^\pi(s, \mathbf{u})$ and state value function $V^\pi(s)$ are used to evaluate joint policy. The advantage function is $A^\pi(s, \mathbf{u}) = Q^\pi(s, \mathbf{u}) - V^\pi(s)$. For clarity, symbols in bold are used to denote the joint variable of group agents.

In single-agent policy optimization problems(Schulman et al., 2015a), the objective is to maximize the expected action state value function $E_{\pi_\theta}[Q_{\pi_\theta}]$. Similarly, for MAPO with CTDE structure, each agent optimize its local policy individually with estimated $Q$ values from centralized critic. Under this circumstance, the overall objective is

$$for\ agent\ a = 1\ to\ n :$$
$$\max_{\theta_a} E_{(\pi_{\theta_a}, \pi^{-a})} \left[ Q_{(\pi_{\theta_a}, \pi^{-a})}^a \right] \tag{2}$$

Where $Q$ values can be substituted by advantages to reduce the variance.

## 4 APPROXIMATIVELY SYNCHRONOUS ADVANTAGE ESTIMATION IN MULTI-AGENT SYSTEM

In this section, we first introduce marginal advantage estimation which expands advantage functions of SARL to MARL as well to realize credit assignment. And then, we describe how to realize approximatively synchronous advantage estimation based on the marginal advantage function in MAPO problem.

### 4.1 MARGINAL ADVANTAGE ESTIMATION

In this subsection, we are going to solve the challenge of credit assignment through the proposed marginal advantage estimation. We first consider an counter-factual way where advantages are estimated asynchronously but policies are updated synchronously, as shown in figure 1(b). In this case, a counter-factual scene can be defined as: at certain state, for agent $a$, other agent always take fixed actions. In partially observable, full cooperative multi-agent settings, the counter-factual advantage of agent $a$'s action $u^a$ under state $s$ is derived based on the joint action's value(or joint $Q$ value) function $Q(s, \mathbf{u})$

$$A^a(s, \mathbf{u}) = A^a(s, (u^a, u^{-a}))$$
$$= Q(s, \mathbf{u}) - \int_{u^a} Q(s, u^{-a}, u^a)\, d\pi^a(u^a|\tau^a) \tag{3}$$

From the view of agent $a$, the counter-factual advantage depends on other agents' joint action $u^{-a}$, which is a random variable and $u^{-a} \sim \pi^{-a}$. In order to remove the dependency, the marginal $Q$ value function of agent $a$ is defined as

$$Q^a(s, u^a) = E_{u^{-a} \sim \pi^{-a}} \left[ Q(s, (u^a, u^{-a})) \right] \tag{4}$$

Notice that in CTED structure, policy $\pi^a(u^a|\tau^a)$ and $\pi^{-a}(u^{-a}|\tau^{-a})$ are independent. By replacing joint $Q$ value function with marginal $Q$ value function, the marginal advantage function is derived

$$
\begin{aligned}
A^a(s, u^a) &= Q^a(s, u^a) - \int_{u^a} Q^a(s, u^a) \, d\pi^a(u^a|\tau^a) \\
&= \int_{u^{-a}} Q(s, u^a, u^{-a}) d\pi^{-a}(u^{-a}|\tau^{-a}) - \int_{u^a} \int_{u^{-a}} Q(s, u^a, u^{-a}) \, d\pi^{-a}(u^{-a}|\tau^{-a}) d\pi^a(u^a|\tau^a) \\
&= \int_{u^{-a}} \left[ Q(s, u^a, u^{-a}) - \int_{u^a} Q(s, u^a, u^{-a}) \, d\pi^a(u^a|\tau^a) \right] d\pi^{-a}(u^{-a}|\tau^{-a}) \\
&= \int_{u^{-a}} A^a(s, \mathbf{u}) \, d\pi^{-a}(u^{-a}|\tau^{-a})
\end{aligned}
$$

(5)

Such replacement will not change the result of advantage estimation because the substitution of joint $Q$ value is its expectation. Form equation(5), for different agent, the value of marginal advantage is different, which realizes credit assignment. It can be easily proved that if counter-factual advantage $A^a(s, \mathbf{u})$ is an unbiased estimation of joint $Q$ value $Q(s, \mathbf{u})$, then marginal advantage is also an unbiased estimation of marginal $Q$ value(Appendix I).

In a counter-factual scene, from the view of agent $a$, other agents and their fix joint actions $u^{-a}$ can be regarded as part of the environment. Let $(s, \mathbf{u}^{-a}) = s_{ctf}$ and counter-factual advantage function can be written as

$$
\begin{aligned}
A^a(s, \mathbf{u}) &= A^a(s_{ctf}, u^a) \\
&= Q(s_{ctf}, u^a) - \int_{u^a} Q(s_{ctf}, u^a) \, d\pi^a(u^a|\tau^a)
\end{aligned}
$$

(6)

In counter-factual scenes, counter-factual advantage function is identical to advantage function in SARL, which means the counter-factual advantage in equation(5) can be replaced by any form of advantage function used in SARL. For example, considering using TD residual $\delta_t^a = r(s_t, u_t^a) + \gamma V(s_{t+1}) - V(s_t)$ as an estimation of joint advantage $A^a(s_t, \mathbf{u}_t)$, the marginal advantages could be written as

$$
\begin{aligned}
A^a(s_t, u_t) &:= E_{u^{-a} \sim \pi^{-a}} \left[ \sum_{l=0}^{\infty} \gamma^l \delta_{t+l}^a \right] \\
A^a(s_t, u_t) &:= E_{u^{-a} \sim \pi^{-a}} \left[ \delta_t^a \right]
\end{aligned}
$$

(7)

The former is unbiased estimation, but has high variance. The latter is biased estimation for any $V \neq V^\pi$, but has much lower variance. These two methods can be combined for compromise between bias and variance(Schulman et al., 2015b).

As agents' policies are independent, the expectation in equation(5) can be split into a $(n-1)$-layer integration, which is complicated. For simplicity and efficiency, the Monte-Carlo(MC) sampling can be applied as a substitution.

$$
A^a(s_t, u_t) = \int_{u^{-a}} A^a(s_t, \mathbf{u}_t) d\pi^{-a} \approx \frac{1}{m} \sum_{u^{-a}}^{m} A^a(s_t, \mathbf{u}_t)
$$

(8)

Where $m$ is the number of other agents' joint action samples. The principle of one step process to calculate marginal advantage with TD residual is shown in figure 2. Firstly, based on the last true state $s_t$, $m$ joint action samples are sampled. These samples are then reorganized. Take agent 1 as example, action $u_{1,t}$ from $S_{a1}$ is combined with other agents' action samples from $S_{a2}$ to $S_{am}$ respectively. As a result, $m$ reorganized new samples are acquired. Based on these new samples, one step simulations are executed and $m$ counter-factual rewards and states are acquired, which are used to calculate the estimation of marginal advantage. At last, the next true state is selected form counter-factual states.

Both methods in equation(7) use $V$ value predictor and require interactive simulation. Agent needs to interact with environment to get extra samples. In this work, we consider using centralized critic to predict joint $Q$ values, and the marginal advantages can be directly calculated with these $Q$ values, which avoids interactive simulation.

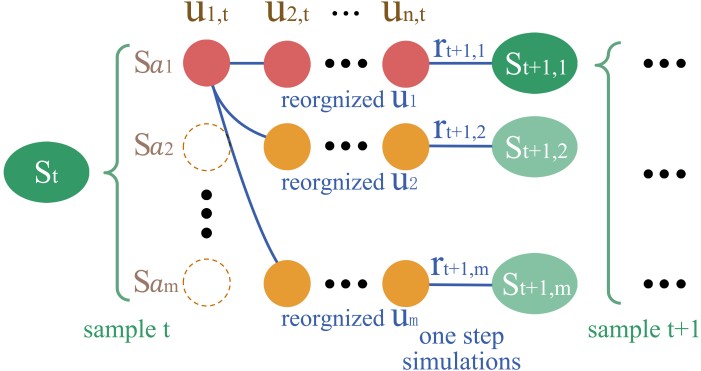

Figure 2: One step process of marginal advantage $A^a(S_t, Sa_1(U_{1,t}))$'s estimation with TD residual. $S_t$ represents true state of time step $t$. $Sa_m$ represents $m$th joint action sample and $U_{n,t}$ denotes the action of agent $n$ in corresponding sample. Blue solid lines connect the reorganized joint action samples. Based on these samples, one step simulations are executed and $S_{t+1,m}$, $r_{t+1,m}$ represent counter-factual state and reward acquired in simulation respectively. Finally, the next true state is selected from these counter-factual states.

## 4.2 Approximatively Synchronous Advantage Estimation

In marginal advantage estimation, actions are sampled from the agents' past policies. The estimation is still asynchronous because it assumes the invariance of others' policies. However, synchronous advantage estimation requires the prediction of other agents' future action, as shown in figure 1(c). In marginal advantage estimation, problem of action prediction becomes policy prediction.

Direct prediction of others' future policies is very difficult. In iterative training, only others' policies of next iteration are needed. Assume others' joint policy of iteration $i$ is $\pi_i^{-a}(u^{-a}|\tau^{-a})$. Synchronous marginal advantage is given by

$$A_{i,syn}^a(s, u^a) = E_{u^{-a} \sim \pi_i^{-a}} \left[ A_i^a(s, u^a, u^{-a}) \right] \tag{9}$$

To calculate the synchronous marginal advantage, we first introduce an approximation that $A_i^a(s, \mathbf{u}) \approx A_{i-1}^a(s, \mathbf{u})$. The reliability of this approximation is ensured by a restriction $KL\left[\pi_i^a, \pi_{i-1}^a\right] < \delta$ Schulman et al. (2015a). For simplicity, we use $\pi^a$ to represent $\pi^a(u^a|\tau)$. In marginal advantage estimation, we have introduced Monte Carlo(MC) sampling and samples form others' joint policy $\pi_i^{-a}$ are needed. However, only polices before iteration $i$ are available. So the second approximation is introduced as $\pi_{i-1}^{-a} \approx \pi_i^{-a}$. Similarly, in order to ensure the approximation is reliable, a KL divergence restriction between $\pi_i^{-a}$ and $\pi_{i-1}^{-a}$ is applied as $KL\left[\pi_i^a, \pi_{i-1}^a\right] < \delta$. The objective of policy optimization problem with synchronous advantage estimation for agent $a$ is

$$\max_{\pi_i^a} E_{u^a \sim \pi_{i-1}^a} \left[ A_{i-1,syn}^a(s, u^a) \cdot \frac{\pi_i^a}{\pi_{i-1}^a} \right]$$
$$= \max_{\pi_i^a} E_{\mathbf{u} \sim \pi_{i-1}^a} \left[ A_{i-1}^a(s, \mathbf{u}) \cdot \frac{\pi_i^a}{\pi_{i-1}^a} \right] \tag{10}$$
$$subject\ to:\ KL\left[\pi_i^{-a}, \pi_{i-1}^{-a}\right] < \delta_1$$
$$KL\left[\pi_i^a, \pi_{i-1}^a\right] < \delta_2$$

The first restriction involves other agents' polices, which requires joint optimization of all agents' policies. The integral objective of multi-agent policy optimization with $n$ agents is

$$\max_{\pi_i^a} \sum_a^n E_{\mathbf{u} \sim \boldsymbol{\pi}_{i-1}} \left[ A_{i-1}^a(s, \mathbf{u}) \cdot \frac{\pi_i^a}{\pi_{i-1}^a} \right]$$

$$subject \ to : \bigcup_a^n KL \left[ \pi_i^{-a}, \pi_{i-1}^{-a} \right] < \delta_1 \tag{11}$$

$$\bigcup_a^n KL \left[ \pi_i^a, \pi_{i-1}^a \right] < \delta_2$$

It can be proved that $KL \left[ \pi_i^{-a}, \pi_{i-1}^{-a} \right] < \sum_o^{-a} KL \left[ \pi_i^o, \pi_{i-1}^o \right]$(Appendix II). For simplification, a tighter form of the restriction $KL \left[ \pi_i^{-a}, \pi_{i-1}^{-a} \right] < \delta_1$ can be written as

$$KL \left[ \pi_i^o, \pi_{i-1}^o \right] < \frac{\delta_1}{n-1} = \delta_1', \ for \ o \ in \ \bigcup^{-a} \tag{12}$$

By replacing the restriction $KL \left[ \pi_i^{-a}, \pi_{i-1}^{-a} \right] < \delta_1$ with the tighter form, the first restriction in equation(11) is simplified:

$$\bigcup_a^n \bigcup_o^{-a} \{ KL \left[ \pi_i^o, \pi_{i-1}^o \right] < \delta_1' \}_a \tag{13}$$

Notice that there are $n-1$ duplicate restrictions for each $KL \left[ \pi_i^a, \pi_{i-1}^a \right] < \delta'$, remove redundant duplicates and the first restrictions in equation(11) finally equals to

$$\bigcup_a^n KL \left[ \pi_i^a, \pi_{i-1}^a \right] < \delta_1' \tag{14}$$

Set $\delta_1 = (n-1)\delta_1' = (n-1)\delta_2$ and the two restrictions in equation(11) can be combined into $\bigcup_a^n KL \left[ \pi_i^a, \pi_{i-1}^a \right] < \delta_2$.

The integral problem of MAPO in equation(11) consists of $n$ individual policy optimization problems with $n$ sub-restrictions. In CTED structure, policies of different agents are updated independently. For agent $a$, only the sub-restriction $KL \left[ \pi_i^a, \pi_{i-1}^a \right] < \delta_2$ is effective. Thus, for further simplification, the integral objective can be split into $n$ sub-objectives:

$$for \ a \ in \ 1, 2, \cdots, n :$$

$$\max_{\pi_i^a} E_{\mathbf{u} \sim \boldsymbol{\pi}_{i-1}} \left[ A_{i-1}^a(s, \mathbf{u}) \cdot \frac{\pi_i^a}{\pi_{i-1}^a} \right] \tag{15}$$

$$subject \ to : \ KL \left[ \pi_i^a, \pi_{i-1}^a \right] < \delta_2$$

The sub-objectives above are similar to the objective in trust region policy optimization problemSchulman et al. (2015a). It's proved that the KL divergence restriction can be effectively replaced by a clip operation(Schulman et al., 2017). The sub-objectives of MAPO with ASAE is finally acquired

$$for \ a \ in \ 1, 2, \cdots, n :$$

$$\max_{\pi_i^a} \sum_1^m \left[ A_{i-1}^a(s, \mathbf{u}) \cdot clip(\frac{\pi_i^a}{\pi_{i-1}^a}, 1 - \epsilon, 1 + \epsilon) \right] \tag{16}$$

## 5 EXPERIMENTS

In this section, we use COMA advantage as counter-factual advantage to estimate the approximatively synchronous advantage. And we compare our method with baseline algorithms on the benchmark StarCraft multi-agent challenge(SMAC).

## 5.1 EXPERIMENT SETTINGS

StarCraft II is a Real Time Strategy(RTS) game. And SMAC is a popular benchmark for cooperative MARL algorithms which provides an interface for RL agents to interact with StarCraft II, getting rewards, observations and sending actions. In our experiments, we consider different types of tasks of battle games involving both mixed and single type of agents. Specifically, our experiments are carried out on 8 tasks of different difficulty level, as shown in table 1. In, homogeneous tasks, agents are of the same type. In symmetric battle scenarios, each army are composed of the same units, agents need to learn to focus fire without overkill. The asymmetric scenarios are more challenging because the enemy army always outnumbers allied army by one or more units. In micro-trick tasks, agents' are required a higher-level of cooperation and a specific micromanagement trick to defeat the enemy, which is the most difficult.

Table 1: Information of SMAC tasks in experiments

| name | Ally Units | Enemy Units | Type |
|---|---|---|---|
| 3m | 3 Marines | 3 Marines | homogeneous&symmetric |
| 8m | 8 Marines | 8 Marines | homogeneous&symmetric |
| 3s5z | 3 Stalkers 5 Zealots | 3 Stalkers 5 Zealots | heterogeneous&symmetric |
| 1c3s5z | 1 Colossi 3 Stalkers 5 Zealots | 1 Colossi 3 Stalkers 5 Zealots | heterogeneous&symmetric |
| 2m_vs_1z | 2 Marines | 1 Zealot | micro-trick |
| 10m_vs_11m | 10 Marines | 11 Marines | homogeneous&asymmetric |
| 2c_vs_64zg | 2 Colossi | 64 Zerglings | micro-trick |
| MMM2 | 1 Medivac 2 Marauders 7 Marines | 1 Medivac 3 Marauders 8 Marines | heterogeneous&asymmetric |

In our experiment settings, only ally units are considered to be MARL agents. The environment is set to be partially observable and each agent has a sight range, which is set to be a circular area around the agent. Only specific attributes of units in sight range are observable. These attributes include distance, relative $x$ coordinate, relative $y$ coordinate, health and shield. Agents can also observe the terrain features surrounding them. And in addition, the last actions of ally units are accessible for all agents. The global state including information about all units on the map and it's only available in centralized training. The action space is discrete and consists of 4 moving directions, $k$ attack actions where $k$ is the number of the enemy units in map, stop and none-operation. The SMAC environment provides both sparse reward and shaped reward settings, we only consider the shaped reward situation where reward are much denser.

In ASAE, we use COMA advantage as counter-factual advantage to calculate the approximatively synchronous advantage. We adopted the CTED architecture and the structure of actors and critic in our methods is the same to other policy based MARL methods, such as COMA. The centralized critic is used to predict the joint $Q$ value function of reorganized samples. The state value is calculated with the average $Q$ values of these reorganized samples, which avoids interactive simulations. The number of sample $m$ is set to be 50 and the clip range is 0.1.

We compare our method with baseline algorithms including IQL(Tan, 1993), VDN, QMIX and COMA. In these baselines, only COMA is policy based and most similar to our method. For all algorithms, good samples are added to replay buffer in initializing stage for early policy training. And all algorithms are trained for 3 million time-steps. The training batch-size is set to be 32 and discount factor is 0.99. The learning rates for critic and actors are both 0.0005.

## 5.2 EXPERIMENT RESULTS

The test wining rates during training are shown in figure 3. Compared to other baseline algorithms, our algorithms converges fastest and perform best in most of the tasks. Particularly, compared to

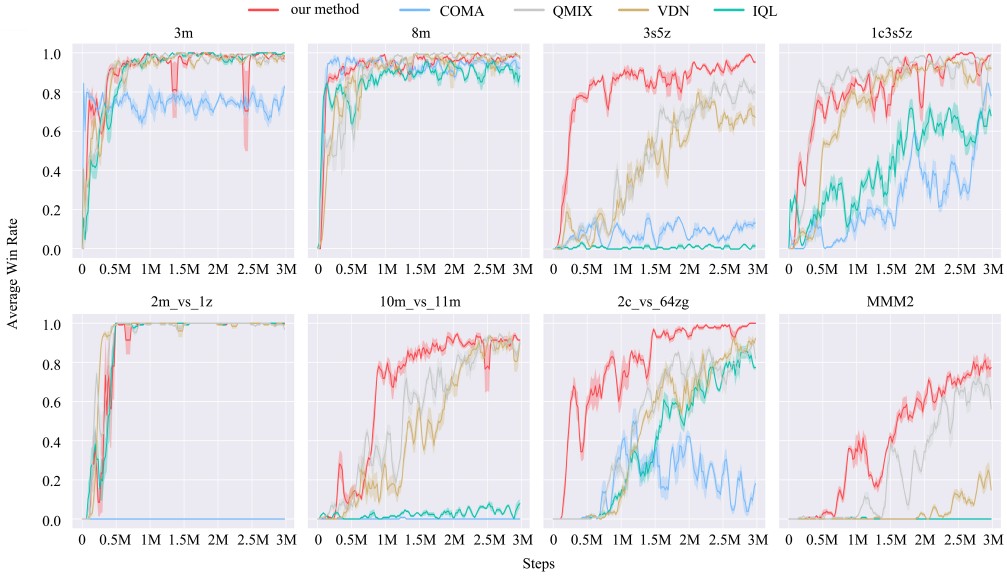

Figure 3: Test wining rate vs. training steps of various methods on SMAC benchmarks

the other policy based MARL algorithm COMA, our algorithm shows considerable improvement in 7 tasks of 8. According to the results, in homogeneous & symmetric tasks such as $3m$ and $8m$, our algorithm converges after 1 million steps of training and reach approximate 100 percent test wining rate. For homogeneous & asymmetric tasks($10m\_vs\_11m$) and simple heterogeneous & symmetric tasks such as $3s5z$ and $1c3s5z$, our algorithms converges slower, and the wining rate fluctuates slightly during training. However, our algorithm also reaches approximate 100 percent test win rate after 3 million steps of training. For different micro-trick tasks, the performance and convergence speed of our algorithm varies greatly. While in harder tasks as $10m\_vs\_11m$, $MMM2$ and $2m\_vs\_1z$, COMA algorithm shows no performance. The wining rate after training is tested and shown in table 2. Our algorithm also shows the best performance in most of the tasks.

An interesting phenomenon is that, the wining rate curve shows less fluctuation in tasks with homogeneous ally units, such as $3m$, $2m\_vs\_64zg$, $2m\_vs\_1z$, etc. It's inferred that, in such tasks, different agents are functionally replaceable, which provides a higher fault tolerance rate for individual agent's action. As a result, the performance fluctuation of certain agent during training has less influence on group's joint policy.

In order to analyse the cooperative strategies learned by agents, we render the battle process between default AI and agents trained by our algorithms. Some key frames are showed in figure 4. Agents in red are ally units. In The first task $2s3z$, cooperative agents learn to focus fire after training.

Table 2: Test wining rates

| ENV | Algorithms | | | | |
|---|---|---|---|---|---|
| | ours | COMA | QMIX | IQL | VDN |
| 3m | **1.0** | 0.81 | **1.0** | **1.0** | **1.0** |
| 8m | 0.97 | 0.97 | **1.0** | 0.91 | **1.0** |
| 3s5z | **0.95** | 0.2 | 0.84 | 0 | 0.72 |
| 1c3s5z | **1.0** | 0.83 | 0.97 | 0.72 | 0.94 |
| 2m_vs_1z | **1.0** | 0 | **1.0** | **1.0** | **1.0** |
| 10m_vs_11m | **0.98** | 0 | 0.97 | 0.31 | 0.97 |
| 2c_vs_64zg | **0.97** | 0.25 | 0.88 | 0.78 | 0.91 |
| MMM2 | **0.78** | 0 | 0.72 | 0.03 | 0 |

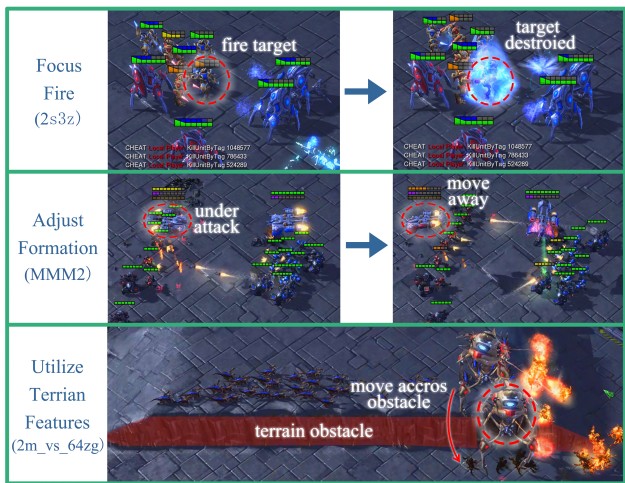

Figure 4: Display of learned cooperative strategies.

While the enemy agents tend to attack the units nearby. After few rounds of crossfire, enemy group quickly lose the first unit. In the second task $MMM2$, besides focus fire, cooperative agents also learn to adjust formation and use skill to avoid being destroyed. Particularly, in micro-trick tasks, cooperative agents learn to take advantage of map features and units' differences. As shown in the third sub-graph, in task $2m\_vs\_64zg$, only ally units are able to move across terrain. Take advantage of this, ally units can attack enemy and move across the terrain when enemy approaching thus avoid being attacked.

## 6 CONCLUSION

In this work, we propose a novel method of advantage estimation which address credit assignment and synchronous estimation in cooperative multi-agent systems. By introducing marginal $Q$ value function, we derived the marginal advantage function and it's relationship with counter-factual advantage function. Then, we define the counter-factual scene where counter-factual advantage can be replaced by single agent advantage. So we acquire a method for single agent advantage function expanding to multi-agent systems. Based on marginal advantage, we propose the approximatively synchronous advantage estimation. Through policy approximation and constrain simplification, the problem of MAPO is decomposed into multiple sub-problems of SA policy optimization and finally solved by PPO method. Our algorithms are evaluated on battle tasks of SMAC benchmark. Compared to baselines, our algorithms perform best in both training and testing. Moreover, visualized battle processes show that our agents acquire heuristic cooperation strategies after training.

For future work, we are interested in applying our algorithm to cooperative robot tasks. For example, two arms' cooperation tasks where two arms are treated as individual agents and need to cooperate to accomplish complicate work. Moreover, because our method provides a channel for SA advantage function expanding to multi-agent system, it's also interesting to investigate the application of various policy based RL algorithms on multi-agent scenarios.

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

## A  APPENDIX I

Assume that joint advantage $A^a(s, \mathbf{u})$ is a unbiased estimation of joint Q value function $Q(s, \mathbf{u})$. Then

$$A^a(s, \mathbf{u}) = Q(s, \mathbf{u}) - b_s$$
$$where : \nabla_\theta b_s \equiv 0 \tag{17}$$

Then

$$
\begin{aligned}
A^a(s, u) &= E_\pi^{-a} \left[ A^a(s, \mathbf{u}) \right] \\
&= E_\pi^{-a} \left[ Q(s, \mathbf{u}) - b_s \right] \\
&= E_\pi^{-a} \left[ Q(s, \mathbf{u}) \right] - E_\pi^{-a} \left[ b_s \right]
\end{aligned} \tag{18}
$$

$E_\pi^{-a} \left[ Q(s, \mathbf{u}) \right]$ is exact the marginal Q value function and $\nabla_\theta E_\pi^{-a} \left[ b_s \right] = E_\pi^{-a} \left[ \nabla_\theta b_s \right] \equiv 0$

## B  APPENDIX II

Consider two agents, whose policies of episode $i$ are represented by $\pi_i^1$ and $\pi_i^2$ respectively.

$$
\begin{aligned}
KL \left[ \pi_i^1 \pi_i^2, \pi_{i-1}^1 \pi_{i-1}^2 \right] &= \int \pi_i^1 \pi_i^2 \log \frac{\pi_i^1 \pi_i^2}{\pi_{i-1}^1 \pi_{i-1}^2} \, du \\
&= \int \pi_i^1 \pi_i^2 \left( \log \frac{\pi_i^1}{\pi_{i-1}^1} + \log \frac{\pi_i^2}{\pi_{i-1}^2} \right) \, du \\
&< \int \pi_i^1 \log \frac{\pi_i^1}{\pi_{i-1}^1} \, du + \int \pi_i^2 \log \frac{\pi_i^2}{\pi_{i-1}^2} \, du \\
&= KL \left[ \pi_i^1, \pi_{i-1}^1 \right] + KL \left[ \pi_i^2, \pi_{i-1}^2 \right]
\end{aligned} \tag{19}
$$

The relation can be expanded to joint distribution of other agents' policies

$$
\begin{aligned}
KL \left[ \pi_i^{-a}, \pi_{i-1}^{-a} \right] &= \int \prod_o^{-a} \pi_i^o \log \frac{\prod_o^{-a} \pi_i^o}{\prod_o^{-a} \pi_{i-1}^o} \, du \\
&< \sum_o^{-a} KL \left[ \pi_i^o, \pi_{i-1}^o \right]
\end{aligned} \tag{20}
$$

