# OpenReview forum: "Multi-agent Policy Optimization with Approximatively Synchronous Advantage Estimation"
_ICLR.cc/2021/Conference — Reject_

### Official Review · AnonReviewer2 · 2020-10-21
**Interesting method, missing ablations, some lack of detail, potentially unfair experimental comparison.**

**Rating:** 5
**Confidence:** 4

**Review:**

This works follows the footsteps of other centralized value function approaches for multi-agent learning, building directly on the counterfactual value function proposed in COMA.

The main idea is to sample additional joint actions to variance-reduce the advantage estimate for each of the agents at each time step.
The paper also introduces a KL penalty that is supposed to ensure that the policies are updated slowly, making the advantage estimate more accurate.

A few comments:
1) ", we consider using centralized critic to predict joint Q values, and the marginal advantages can be directly calculated with these Q values, which avoids interactive simulation.". This directly contradicts Figure 2, where it is mentioned that "one step simulations are executed" for each of the samples. This point is relevant since extra simulation steps would drastically change the sample requirements for the method. Furthermore, this assumes that there is a simulator that can be set to arbitrary state transitions, which is different from the standard RL assumption.
2) "which means the counter-factual advantage in equation(5) can be replaced by any form of advantage function used in SARL.". This is generally true. How to estimate the advantage is orthogonal to single agent vs multi-agent learning. In particular, the COMA critic (Q-function) could easily be replaced with a GAE or similar. On a related note, while using a critic that depends only on the Q(s,u) (ie. the central state and joint action) is common practice in this line of work, it is not generally appropriate. Clearly, in Dec-POMDPs, the future actions will depend on the action-observation histories (AOHs), $\tau$ of all the agents. As a consequence, in general the critic should condition on these AOHs.
3) "And in addition, the last actions of ally units are accessable for all agents." This is different form the standard SMAC setting, where only the last actions of the *visible* agents are observable by each of the agents. Please clarify this rationale.
4) Please specify what advantage estimation is being used in this paper. My understanding is that the paper uses the joint-Q function similar to the COMA paper. Is this accurate?
5) Overall the results look competitive, but are currently not very informative due to missing ablations. It's impossible to tell if the PPO-like KL penalty or the multi-sample variance reduction are responsible for the improved performance.
6) The training curves look very odd. In particular, they look unstable but have very narrow errorbars, indicating that many of the runs drop at the same time. This suggests that the experiments may have accidentally reused the same random seed.
7) MMM2 is the only "superhard" scenario used from SMAC. Did the method fail on the other ones?

---

### Official Review · AnonReviewer4 · 2020-10-27
**Review of Multi-agent Policy Optimization with Approximatively Synchronous Advantage Estimation**

**Rating:** 5
**Confidence:** 2

**Review:**

Summary

The paper deals with the problem of credit assignment and synchronous estimation in cooperative multi-agent reinforcement learning problems. The authors introduce marginal advantage functions and use them for the estimation of the counterfactual advantage function. These functions permit to decompose the Multi-Agent Policy Optimization Problem in Single Agent Policy Optimization subproblems, which are solved using TRPO.


Strengths

The paper proposes elegant derivations to deal with the multi-agent policy optimization problem.  The marginal advantage function gives the possibility to solve the credit assignment problem. Instead, the synchronous advantage estimation assumes that the current policy is close to the previous one. To rely on this approximation, the authors propose to add KL constraints to the optimization problem, which lead to derivations similar to TRPO.
The approach is interesting and the experimental results confirm that the method performs better than COMA.



Weakness

The paper is only experimental and gives no theoretical guarantees.
In the results of the experiments, it seems that QMIX has excellent performance, sometimes better than the proposed method, but in the Experimental Results, there are no comments on this fact. Could you give some insight into why QMIX, in some cases, performs better than the proposed method?
Is it possible to perform an ablation study on the clipping parameter? It seems the most relevant hyperparameter of the proposed method since it regularized the assumption of using the past policy as the current one.


Minor:
- figure 6 reorgnized $\rightarrow$ reorganized
- missing space in second line under equation 9

---

### Official Review · AnonReviewer3 · 2020-10-28
**Unclear presentation and insufficient experimental proof**

**Rating:** 3
**Confidence:** 4

**Review:**

Summary

The authors address the multi-agent learning issue of agents needing to infer other agents' policies to estimate their expected reward correctly. This can be done synchronously (agents see other policy models), or asynchronously (agents cannot see other agents' policies) -- if I understand the authors' nomenclature/illustrations correctly.

Certainly improving the stability of multi-agent learning with just first-order methods and without the need to infer other agent policies is an important and interesting problem. However, I found the manuscript hard to follow and experimental proof unconvincing.

- Clarify issues: It seems that the authors end up with a multi-agent variation of TRPO in Eq 16. They seem to basically assume that agent policies are independent, so they can rewrite the advantage A(s, u^a) of agent a as marginalization of A(s, u^joint) over the actions/policies of the other agents. From this, they then apply TRPO to each agent individually, using the most conservative trust region radius among the N agents. I'm not convinced this is that novel.
- The dashed lines in Figure 1 make the relationships between agents a bit hard to read.
- There are minor English grammar and syntax issues throughout, e.g., ' each agent has a local trajectory τa consists of historical observation and action'.
- The authors should visualize and better explain why in certain situations their algorithm seems to do better or not (e.g., 2c_cs64zg vs 2m_vs_1z: what's the difference? The authors only mention that 2m_vs_1z is "hard").
- The results in Figure 3 seem to use only 1 seed? The authors should repeat experiments with more (5 or more) random seeds to show how significant the gains are.

I encourage the authors to add more experimental proofs and make the presentation clearer.

---

### Official Review · AnonReviewer1 · 2020-10-28
**Two useful algorithmic contributions but writing requires much improvement**

**Rating:** 4
**Confidence:** 4

**Review:**

Summary of claims and contributions:
The paper introduces two improvement for the centralised-learning and decentralised-execution architecture for multiagent reinforcement learning algorithms: i) a per-agent marginal advantage estimator that marginalises other agents' actions out; ii) motivates a trust-region method. The authors claim the former alleviates the credit-assignment problem, while the later reduces bias in the approximation of the advantage function due to assuming other agents' policies stationary.
Numerical experiments show that the proposed algorithm match or outperforms some state of the art baselines.

Strong points:
The marginal advantage estimator is derived in two different ways---first and last line of (5)---, and both make intuitive sense. This estimator seems a natural extension of the counter-factual advantage estimator due to Foerster et al. (2018).

The connection with trust-region methods to reduce bias is also interesting, I personally tend to think on trust-region constraints as a method to reduce variance in the parameter update (due to stochastic gradient), while here it is motivated to reduce bias. It is also practical, since it can be easily implemented by using PPO for every agent.

Simulation results show that the proposed improvements perform well in practice.

Weak points:
The writing requires much improvement in order to be publishable. Notation is unclear if not loose and sometimes incomplete (e.g,, indexes in summations and unions are difficult to follow, policy parameter is never introduced, exponent instead of Cartesian product on the observation and action sets wasn't obvious...) There are multiple typos (mainly in text but also some in maths). The draft must be proof-read for English style and grammar issues. This is the main reason for my rejection.

Many experimental details are missing hindering reproducibility.

The draft is 9 pages long. But the paper format could be much improved such that it fits in 8 pages, so I haven't considered this a deal breaker.

Questions:
1. Why in the centralised-learning decentralised-execution are the agents' policies independent of each other?
2. The concept of the counter-factual scene is not clear. Are the authors assuming the agents behave stationarily? Hence, the trust-region constraint bounds the relaxation on this assumption?
3. First line in Sec. 5 says "we use COMA advantage as counter-factual advantage to estimate the approximately synchronous advantage." What does it mean? Don't the authors use the marginal advantage?

---

### Decision · Program_Chairs · 2021-01-07
**Final Decision**

**Decision:**

Reject

**Comment:**

All reviewers expressed interest in this promising approach, but raised questions that were not addressed by the authors during the discussion period. As concerns raised included insufficient repeats of empirical experiments to draw conclusions and the paper appearing to be in an early draft format, we cannot support acceptance for publication at this time. I strongly encourage the authors to act on the feedback given to improve the paper for a future submission.